# Causal Relationships between Oil Prices and Key Macroeconomic Variables in India

**Kamal P. Upadhyaya [1,*], Raja Nag [2] and Franklin G. Mixon, Jr. [3]**

[1]  Department of Economics and Business Analytics, University of New Haven, West Haven, CT 06516, USA
[2]  Department of Accounting and Financial Studies, New York Institute of Technology,
    Glen Head, NY 11545, USA; dnag@nyit.edu
[3]  Center for Economic Education, Columbus State University, Columbus, GA 31907, USA;
    mixon_franklin@columbusstate.edu
[*]  Correspondence: kupadhyaya@newhaven.edu

**Abstract:** India is among the largest and fastest-growing economies in the world. To continue its growth, energy is and will continue to be one of its most important considerations. With a population of over one billion, India is the third largest consumer of petroleum on the globe. To maintain this ranking, India imports a large percentage of its total oil consumption. Given India's current position as a large importer of oil, how does oil price volatility affect the Indian economy? This paper examines the effect of oil price volatility on inflation, economic growth, and the stock market in India. Statistical tests suggest that the overall price level, the real effective exchange rate, and oil prices are negatively related to aggregate output in the long run. Granger causality test results derived from a vector error correction model support bidirectional causality between oil prices and aggregate output, indicating that a change in oil prices also affects aggregate output in the short run.

**Keywords:** oil prices; aggregate output; inflation; real exchange rates; stock prices; India

**JEL Classification:** C22; E20; E30; F10; F30

## 1. Introduction

India is currently the fifth-largest economy in the world. Even so, it is also one of the fastest-growing economies on the globe. To continue its growth, energy is and will continue to be one of its most important considerations. One of India's primary sources of energy is oil, which is not only used in its manufacturing sector but also in its agriculture and transportation sectors, with the latter focused on the transport of both agricultural and industrial goods. With a population of over one billion, India is the third largest consumer of petroleum on the globe. To maintain this ranking, India imports a substantial percentage of its total oil consumption. The world oil market can be extremely volatile. Given India's current position as a large importer of oil, how does oil price volatility affect the Indian economy?

Despite the importance of the question above, there is a relative dearth of academic research on oil price fluctuations and their effect on macroeconomic variables in India. This paper addresses this gap in the literature by examining the effect of oil price volatility on inflation, economic growth, real exchange rates, and the stock market in India. First, cointegration tests are applied to monthly data from January 2001 to June 2020 in order to test for long-run relationships among the variables listed above. Next, a vector error correction model is estimated in order to obtain Granger causality and variance decomposition information for the variables in the model. Lastly, impulse response functions are provided in order to analyze the effect of oil price shocks on aggregate output, inflation, the real exchange rate, and stock prices.

Empirical results reported below in this study suggest that the overall price level, the real effective exchange rate, and oil prices are negatively related to India's aggregate output in the long run. Moreover, test results relying on a vector error correction model support bidirectional causality between oil prices and aggregate output, indicating that a change in oil prices also affects India's aggregate output in the short run. These test results suggest that innovations in oil prices explain just over 10.5 percent of the variation in India's aggregate output over three months, rising to about 11.5 percent over six months before falling to 10.25 percent over nine months and finally to about 9.5 percent over one year. Relatedly, impulse response functions show that a shock in oil prices impacts aggregate output beginning in the third period, reaching a plateau after the sixth period.

Before turning to more of the specifics of the aforementioned empirical results, Section 2 of this study provides a review of the economics literature that builds a solid theoretical foundation for our study. This review is followed, in Section 3, by a discussion of the study's theoretical foundation, as well as a description of the statistical methodology and data used to test our hypotheses. In Section 4, the estimation and the empirical findings are reported. Finally, Section 5 presents a summary and policy recommendations related to the study.

## 2. Prior Literature: A Review

Volatility in oil prices affects various macroeconomic variables, such as the level of output, inflation, stock prices, and even the balance of payments. The transmission mechanism involving oil price volatility includes both aggregate demand and aggregate supply. The demand side transmission channel emanates from both consumption and investment choices made by households. For example, a change in oil prices directly affects household consumption and purchasing patterns. Given that the demand for oil is relatively price-inelastic, any increase in oil prices will likely induce households to reduce their expenditures on other items. Relatedly, Sill (2007) asserts that an increase in oil prices shifts the demand for other goods leftward because an increase in oil prices simultaneously diminishes wealth and creates uncertainty about the future. Similarly, Fernald and Trehan (2005) assert that a rise in oil prices is akin to a consumption tax, given that the additional payment made by domestic households to foreign oil producers reduces households' purchasing options with regard to all other goods and services. Lastly, Bernanke (1983) suggests that rising oil prices negatively affect short-run economic performance largely because of the delay in, or at times the cancellation of, households' decisions to buy large-ticket items and investment goods.

Given that crude oil is an important intermediate input in various production processes, particularly those resulting in petroleum-based products, any increase in the price of oil potentially impacts a number of macroeconomic indicators. As a result, there is no dearth of empirical studies that explore the effect of oil price fluctuations on various macroeconomic variables (e.g., see Hamilton 1996, 2003; Hooker 1996, 1999; Huntington 1998; Ahmed and Wadud 2011; Marín-Rodriguez et al. 2022). In a seminal piece, Hamilton (1983) classifies oil price shocks as the major contributing factor in nine out of 10 U.S. recessions after World War II. Cebula et al. (2002), Cunado and Gracia (2005), Cologni and Manera (2008), and Kilian (2008) also find similar effects on macroeconomic variables from oil price shocks. Basnet and Upadhyaya (2015) utilize a structural vector autoregressive (SVAR) approach to analyze the impact of oil price shocks on real output, inflation, and the real exchange rate in Thailand, Malaysia, Singapore, the Philippines, and Indonesia (i.e., the ASEAN-5).[1] They find that oil price fluctuations do not affect the ASEAN-5 economies in the long run, nor do they explain a significant variation in the inflation or real exchange rates.[2]

An earlier study by Jiménez-Rodríguez and Sánchez (2005) examines the effect of oil price shocks on the real economic activity of major industrialized countries, which include both oil-exporting and oil-importing countries. Their findings suggest that an increase in oil prices reduces real GDP growth in oil-importing countries, while the effect of oil price shocks on the economies of oil-exporting countries is ambiguous (Jiménez-Rodríguez and

Sánchez 2005). Utilizing a VAR technique, Farzanega and Markwardt (2009) analyze the dynamic relationship between oil price shocks and major macroeconomic variables in Iran, a major oil exporting country. They find a positive relationship between oil price growth and industrial output while also detecting an inflationary effect and evidence of Dutch Disease due to an appreciation of the domestic currency. Brini et al. (2016) conducted a similar study of prominent oil-exporting and oil-importing MENA (Middle East and North Africa) countries using an SVAR model.[3] The impulse response functions they derived suggest that, in the long run, oil price fluctuations affect the real exchange rate in oil-importing countries, most notably Tunisia and Morocco, while their impact on inflation is relatively small, likely as a result of product price subsidies.

In addition to the aforementioned macroeconomic variables, oil prices also impact financial markets. In an early study, Sadorsky (1999) finds that oil price fluctuation explains a large portion of the variation in stock returns. Likewise, Kilian and Park (2009) find that the demand and supply shocks driving the global crude market jointly account for 22 percent of the long-run variation in real stock returns in the U.S. A more recent study by Wei and Guo (2017) on China finds an unstable relationship between oil prices and the stock market. Another study investigating the long-run relationship between crude oil prices and the Chinese stock market, this one by Wei et al. (2019), uses a nonlinear threshold cointegration approach. The findings suggest that Chinese stock prices and oil prices (futures) are cointegrated in the long run despite structural breaks in 2008 and 2012. In addition, the study also finds that oil futures markets have a significant impact on China's stock market, both directly and indirectly, through various macroeconomic channels. Lastly, a recent study on the South Asian stock market that explores the nexus between oil prices and the stock market reveals that higher world oil prices stimulate stock prices (Alamgir and Amin 2021).[4]

Finally, although recent research by Shahabad and Balcilar (2022) examines the dynamic interactions between oil prices (and other financial variables) and economic policy uncertainty in India, finding that the prices of oil, natural gas, and gold are all interconnected with economic policy uncertainty and vice-versa, it does not attempt to link oil price shocks to traditional macroeconomic variables such as real output, inflation, and real exchange rates. Thus, there remains a marked paucity of economic research on oil price shocks and their effect on traditional macroeconomic variables in India. As indicated in the previous section, our study addresses this gap in the literature by examining the effect of oil price volatility on inflation, economic growth, real exchange rates, and the stock market in India. The methodology and data employed to do so are each explained in the next section.

## 3. Methodology and Data

To explore the relationship between the stock prices, oil prices, aggregate output, price level, and the real exchange rate in India, the following specification is introduced,

$$Y_t = f(SENSEX_t, OP_t, CPI_t, RER_t), \tag{1}$$

where $Y_t$ is India's output level at time $t$, as proxied by India's industrial output, $SENSEX_t$ is the value of India's SENSEX stock index at time $t$, $OP_t$ is the price of oil at time $t$, $CPI_t$ is the consumer price index at time $t$, and $RER_t$ is the Indian rupee's real effective exchange rate at time $t$.[5] Rising stock prices generates a wealth effect and an increase in consumer confidence, which then lead to a rise in aggregate demand and economic output. The resulting economic expansion couples with the expectation of future economic growth to boost investment demand (Upadhyaya et al. 2017). This boost in investment demand shifts the demand for stocks rightward, leading to an increase in stock prices. Therefore, we expect a two-way causality between aggregate output, $Y_t$, and stock prices, $SENSEX_t$.

As discussed above, the effect of a change in oil prices on aggregate output travels through both the demand and supply sides of the overall economy. For instance, an increase in the level of income (i.e., aggregate output) increases aggregate demand. This

increase in aggregate demand also increases the demand for oil, as oil is a major input in the production and transportation of goods. As far as the supply side is concerned, an increase in oil prices causes an increase in both manufacturing and transportation costs, which impact both manufacturing and agricultural goods. This impact is seen as a negative shock to aggregate supply. As a result of these interactions, aggregate output, $Y_t$, and oil prices, $OP_t$, are expected to exhibit a two-way causality.

The relationship between the inflation rate and real output is not straightforward. An increase in the general price level can induce people to spend more as a hedge against future inflation. This, in turn, can increase the aggregate output level in the short run. However, this can be true only in the case of a high inflationary scenario, not in a low or moderate inflationary environment. In an emerging economy such as India's, a higher inflation rate tends to reallocate resources toward unproductive activities, such as purchases of land, gold, and other ornaments, which inhibits economic efficiency and leads to a decline in aggregate output. Standard macroeconomic theory suggests that an increase in the output level raises the price level and vice-versa. Therefore, one would expect a two-way causality between output and price levels.

Keynesian economics suggests that a depreciation (devaluation) of the domestic currency increases aggregate demand by increasing net exports. Although this has an expansionary effect, currency depreciation can also increase the price of needed inputs that are imported. This latter effect can have a negative impact on aggregate supply and, thus, aggregate output (Upadhyaya and Upadhyay 1999). An increase in aggregate output increases the demand for imports and foreign currencies, which will depreciate the domestic currency. These relationships suggest that a two-way causality exists between the output level, $Y_t$, and the real exchange rate, $RER_t$.

Finally, the theoretical relationships between the variables discussed above suggest that there is interdependence in all of the macroeconomic variables. To account for this, we employ an unrestricted VAR model in order to capture the expected relationships between these variables. We use monthly data from January 2001 to June 2020 (i.e., $n = 240$). With the exception of data from India's SENSEX stock index, which were collected from CNBC, all of the data were collected from the Federal Reserve Bank of St. Louis' website. As stated above, given that the monthly data for real GDP are not available, the Industrial Production Index (IPI) is used. The global price of oil price is measured using the Brent-Europe crude oil price in current U.S. dollars. Lastly, all of the variables are converted into natural logs.

## 4. Estimation and Empirical Findings

### 4.1. Data Series Stationarity

An augmented Dickey–Fuller test (Said and Dickey 1984) and a Phillips–Perron test (Phillips and Perron 1988) were conducted to ensure that the data series are stationary. The results of these tests are reported in Table 1. The test results suggest that all of the data series are stationary in the first differences.

**Table 1.** Unit root test results.

| Variable | Augmented Dickey-Fuller Test | | Phillips-Perron Test | |
|:---:|:---:|:---:|:---:|:---:|
| | Level | First-Diff | Level | First-Diff |
| $Y$ | −1.89 | −12.14 *** | −1.78 | −12.74 *** |
| *SENSEX* | −0.67 | −14.55 *** | −0.74 | −14.61 *** |
| *OP* | −2.33 | −9.67 *** | −2.16 | −11.89 *** |
| *RER* | −0.28 | −7.74 *** | −0.01 | −12.61 *** |

Note: *** denotes the 0.01 level of significance. Key: $Y$ = aggregate output; *SENSEX* = value of the SENSEX stock index; *OP* = price of oil; *RER* = real exchange rate.

### 4.2. Johansen's Cointegration Test

Next, Johansen's cointegration test (Johansen 1988, 1991; Johansen and Juselius 1990) is conducted in order to test for a long-run relationship among the variables. The optimal lag length was determined by the AIC criterion. Johansen's cointegration test results are reported in Table 2. The long-run relationships between the output and other variables are derived after normalizing the coefficient of $Y$ to one, which comes out as follows,

$$Y = 0.46SENSEX^{***} - 0.18OP^{***} - 0.54CPI^{***} - 2.06RER^{***}$$
$$(0.08)\ (0.05)\ (0.12)\ (0.45) \tag{2}$$

where the figures in parentheses are the standard errors of the respective coefficient estimates, each of which is significant at the 0.01 level (as denoted by ***).

**Table 2.** Johansen's cointegration test results.

| $H_0$ | Trace Statistics | 5% Critical Value |
|---|---|---|
| r = 0 | 71.86 ** | 69.89 |
| r ≤ 1 | 38.03 | 47.86 |
| r ≤ 2 | 21.06 | 29.80 |
| r ≤ 3 | 7.60 | 15.50 |
| r ≤ 4 | 0.55 | 3.84 |

Note: ** denotes the 0.05 level of significance.

In light of the fact that all the data series are in natural logarithmic form, the coefficient estimates of the variables can be interpreted as the long-run elasticity with respect to the dependent variable. The negatively signed and statistically significant coefficient attached to the price level, *CPI*, in (2) above indicates that a one percent increase in the general price level lowers aggregate output in India by 0.54 percent in the long run. This finding is consistent with Mahaddes and Raissi (2014). Next, an increase in the real effective exchange rate is associated with lower aggregate output in India in the long run. Indeed, an increase in the real effective exchange rate makes India's exports less competitive, while imports into India become more competitive, leading to a decline in domestic output. In terms of the stock market, the positively signed and significant coefficient estimate attached to the *SENSEX* in (2) above indicates that increasing stock prices has a positive and significant effect on real output in the long run. This finding is consistent with the notion that an increase (or decrease) in stock prices works to increase (or decrease) overall economic activity, corporate profits, and expected future cash flows, which in turn increases (decreases) stock prices and aggregate output (Upadhyaya et al. 2018).[6] Finally, the coefficient estimate attached to *OP* is negative and statistically significant, indicating that an increase in oil prices negatively affects aggregate output in the long run in an oil-importing country such as India. This result is quite consistent with the theoretical expectation discussed above.

### 4.3. Vector Error Correction Model

Given that the cointegration test results suggest that the variables in (1) are cointegrated, following Engle and Granger (1987), a vector error correction model (VECM) is estimated. The VECM estimation also provides Granger causality information for the variables in the model, which is useful for understanding the causal relationships among the variables in the model. The VECM-estimated Granger causality results are reported in Table 3.

The Granger causality results reported in Table 3 suggest that increases in stock prices (*SENSEX*) Granger cause an increase in aggregate output. Indeed, an increase in stock prices creates a wealth effect, leading to an increase in aggregate demand, which in turn increases aggregate output. At the same time, aggregate output also Granger causes stock

prices. In essence, a two-way causality is detected between *Y* and the *SENSEX*.[7] Likewise, a two-way causality is detected between aggregate output and oil prices, as indicated in Table 3. Although the overall price level is Granger caused by oil prices, the price level Granger causes oil prices at only the 0.20 level of significance.[8] Further, the results in Table 3 suggest that aggregate output and the price level both Granger cause the real exchange rate. However, a reverse causality is not detected in either case. The results in Table 3 also suggest that bi-directional causality exists between the *SENSEX* and *RER*.[9] Lastly, the results indicate that, with the exception of the price level, all of the variables jointly Granger cause each of the individual variables.

**Table 3.** Granger causality/Wald test results.

| Dependent Variable | $\Delta Y$ | $\Delta SENSEX$ | $\Delta OP$ | $\Delta CPI$ | $\Delta RER$ | All |
|---|---|---|---|---|---|---|
| $\Delta Y$ | — | 8.69 * | 40.08 *** | 2.25 | 3.05 | 52.27 *** |
| $\Delta SENSEX$ | 7.84 * | — | 11.35 ** | 6.30 † | 27.27 *** | 43.76 *** |
| $\Delta OP$ | 10.52 ** | 4.25 | — | 6.18 † | 4.10 | 24.67 ** |
| $\Delta CPI$ | 2.89 | 5.10 | 10.85 ** | — | 2.97 | 19.27 |
| $\Delta RER$ | 9.41 ** | 10.80** | 3.85 | 15.54 *** | — | 38.47 *** |

Note: The numbers above are $\chi^2$ statistics, where ***(**)[*]{†} denote the 0.01(0.05)[0.10]{0.20} level of significance. Key: *Y* = aggregate output; *SENSEX* = value of the SENSEX stock index; *OP* = price of oil; *CPI* = price level; *RER* = real exchange rate.

*4.4. Variance Decompositions*

In addition to causality, the VECM provides estimates of how a change in each of the variables in the model impacts all other variables in the model. The variance decompositions reveal the proportion of forecast error variance for each variable accounted for by each variable's own innovation and shocks to the other variables in the system (Upadhyaya et al. 2017). The transmission of innovation among variables may occur via many channels. Table 4 presents the variance decompositions for three, six, nine, and 12 months explained by innovation in each of the variables separately (i.e., one at a time). The estimates indicate that over a three-month period, almost 87 percent of the variation in aggregate output, *Y*, is explained by its own innovation. This declines to 82 percent over six and nine months and to 81 percent over 12 months. Among other variables, innovations in oil prices, *OP*, stock prices, and *SENSEX* seem to have some effect in explaining aggregate output. For example, innovation in *OP* explain about 10.66 percent of the variation in *Y* over three months. This changes to 11.49 percent over six months, 10.25 percent over nine months, and finally to 9.46 percent over 12 months. The effect on *Y* of innovation in the *SENSEX* increases from 1.88 percent over three months to five percent over six months and finally to 8.42 percent over 12 months. Lastly, the effect of innovations on the price level, *CPI*, real exchange rate, and *RER* are concerning as they appear to be very small and are statistically insignificant.

Another variable of interest is the price level, *CPI*. Most of its variation is explained by its own innovation. However, the output, real exchange rate, and oil prices also seem to explain portions of its variation. For instance, innovation in *Y* explains about one percent of the variation in *CPI* over three months. This increases to 4.75 percent by 12 months. Likewise, innovation in *RER* explains 3.65 percent of the variation in the *CPI* over six months, which increases to 7.2 percent in period 9 and again to 8.9 percent in period 12. As far as the effect of innovation on the oil price, *OP*, is concerned, it is not relatively large, but it is quite visible. Although its impact is less than 0.5 percent over three months, it increases to 2.12 percent over six months but then declines to 1.61 percent and 1.2 percent, respectively, over nine and 12 months.

One of the purposes of this study is to examine the behavior of the stock market with respect to a change in other variables. The variance decomposition in Table 4 shows that innovation in the output level affects the stock market, *SENSEX*, by six, seven, seven, and six percent, respectively, in periods three, six, nine, and 12. This finding suggests that innovation in the *SENSEX* affects *Y* but also that innovation in *Y* affects the *SENSEX*. This result is corroborated by the findings of the two-way causality between these two variables.

Innovations in other variables, such as *RER* and *OP*, also seem to explain the variation in the *SENSEX*, although the price level, *CPI*, seems to have a relatively small impact. Next, most of the variation in the real exchange rate comes from its own innovation, while other variables affecting it include aggregate output and the price level. Innovation in the aggregate output, for example, explains two percent over three months, about seven percent over six months, and about five and four percent, respectively, over nine and 12 months. Variations in oil prices, *OP*, are significantly explained by innovations in aggregate output, *Y*, and stock prices, *SENSEX*. Given that India represents a large market for oil, this finding is not surprising. It is interesting to note that innovation in oil prices has a significant effect on the variation in aggregate output (i.e., 10.66 percent over three months, 11.49 percent over six months, eventually dropping to 9.46 percent over 12 months), while at the same time innovation in aggregate output also shows a visible effect on the variation in oil prices. Again, this result is corroborated by the two-way causality between these two variables.

**Table 4.** Variance decompositions.

| Period | Y | SENSEX | OP | CPI | RER |
|---|---|---|---|---|---|
| | | Variance decomposition of *Y* explained by | | | |
| 3 | 86.58 | 1.88 | 10.66 | 0.57 | 0.31 |
| 6 | 82.33 | 5.00 | 11.49 | 0.70 | 0.48 |
| 9 | 82.77 | 6.63 | 10.05 | 0.48 | 0.39 |
| 12 | 81.37 | 8.42 | 9.46 | 0.36 | 0.38 |
| | | Variance decomposition of *SSEX* explained by | | | |
| 3 | 6.07 | 85.81 | 2.74 | 0.05 | 5.32 |
| 6 | 7.16 | 86.14 | 2.46 | 0.15 | 4.09 |
| 9 | 6.81 | 87.42 | 1.75 | 0.32 | 3.69 |
| 12 | 6.35 | 88.31 | 1.40 | 0.34 | 3.60 |
| | | Variance decomposition of *OP* explained by | | | |
| 3 | 9.20 | 7.32 | 82.71 | 0.04 | 0.72 |
| 6 | 10.72 | 10.81 | 74.23 | 1.74 | 2.49 |
| 9 | 10.24 | 15.56 | 68.81 | 2.85 | 2.70 |
| 12 | 9.88 | 16.89 | 67.76 | 2.88 | 2.68 |
| | | Variance decomposition of *CPI* explained by | | | |
| 3 | 0.97 | 0.18 | 0.46 | 97.60 | 0.79 |
| 6 | 1.52 | 0.13 | 2.12 | 92.58 | 3.65 |
| 9 | 2.97 | 0.29 | 1.61 | 87.92 | 7.20 |
| 12 | 4.75 | 0.82 | 1.20 | 84.32 | 8.90 |
| | | Variance decomposition of *RER* explained by | | | |
| 3 | 2.18 | 0.18 | 0.04 | 2.47 | 95.12 |
| 6 | 6.61 | 1.07 | 0.54 | 2.79 | 88.99 |
| 9 | 5.53 | 6.20 | 2.95 | 2.54 | 82.76 |
| 12 | 4.37 | 12.20 | 5.41 | 2.36 | 75.66 |

Key: *Y* = aggregate output; *SENSEX* = value of the SENSEX stock index; *OP* = price of oil; *CPI* = price level; *RER* = real exchange rate.

### 4.5. Impulse Response Functions

Finally, impulse response functions are provided in order to analyze the effect of oil price shocks on aggregate output, inflation, the real exchange rate, and stock prices. The impulse response functions are reported in Figure 1. The first row in Figure 1 provides the impulse responses of oil price shocks on its own price, aggregate output, inflation, the real exchange rate, and stock prices, respectively. A shock in oil prices appears to impact

aggregate output beginning in the third period and then reaches a plateau after the sixth period. The impulse response functions indicate little effect on inflation from oil price shocks. The effect of oil price shocks on the real exchange rate begins in period five, while the effect of an oil price shock on stock prices is initially negative but eventually becomes positive by period six. This finding is consistent with Alamgir and Amin (2021), who examined the South Asian stock market.

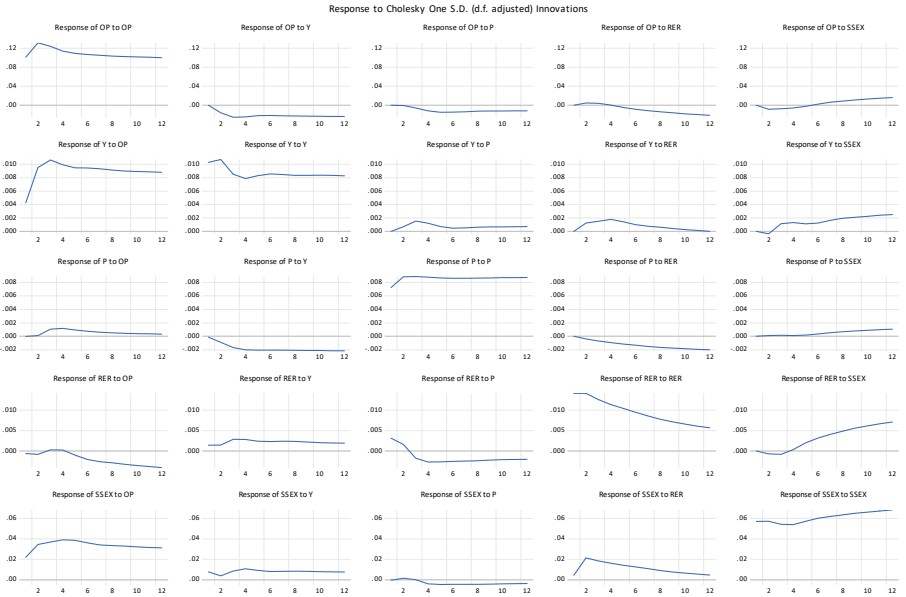

**Figure 1.** Impulse response functions.

## 5. Summary and Conclusions

This paper studies the effect of oil price fluctuations on different macroeconomic variables in India by applying the VECM estimation to monthly data from 2001 to 2022. The empirical results suggest that the price level, the real effective exchange rate, and oil prices are negatively related to aggregate output in the long run. Granger causality test results derived from the VECM estimation support bidirectional causality between oil prices and aggregate output, indicating that a change in oil prices also affects aggregate output in the short run. These results are generally corroborated by variance decompositions and impulse response functions. Finally, Granger causality tests and variance decompositions also indicate that oil prices are significantly related to both the overall price level and stock prices.

The overall VECM estimations that include the variance decompositions and the impulse response functions indicate that oil price fluctuations do impact the macroeconomic variables in India. Although these impacts may appear to be relatively small in the short run, over the long run, as the Indian economy continues industrializing, its dependence on energy will continue to grow. At present, India meets about three-fourths of its daily oil consumption from imports, and its dependence on oil imports is likely to increase even as oil prices continue to rise. Therefore, if appropriate policy measures are not adopted, any crisis that increases the price of oil will likely have destabilizing macroeconomic effects in India. In order to avoid this potential outcome, the Indian government may have to consider tapping all possible energy sources in the country, including its five billion barrels of oil (approximately) currently held in reserve.

Finally, the results presented in this study may not apply globally or even to other South Asian countries. Thus, the approach used here for the study of the impact of oil price shocks on India's economy should be applied to other countries, particularly those in South Asia (e.g., Sri Lanka, Nepal, etc.). Of course, the methods used above for India can also be applied to countries elsewhere on the globe, such as those in Eastern Europe and

South America. This line of future research would be particularly beneficial if applied to countries and areas that have been understudied in prior research.

**Author Contributions:** Conceptualization, K.P.U., R.N. and F.G.M.J.; methodology, K.P.U. and R.N.; data curation, K.P.U. and R.N.; writing–original draft, K.P.U. and F.G.M.J.; writing–review and editing, F.G.M.J.; supervision, K.P.U. and R.N.; administration, K.P.U. and F.G.M.J. All authors have read and agreed to the published version of the manuscript.

**Funding:** This research received no external funding.

**Informed Consent Statement:** Not applicable.

**Data Availability Statement:** Data are available from the authors upon request.

**Acknowledgments:** The authors thank four anonymous reviewers of this journal for helpful comments on a previous draft. Any remaining errors or omissions are our own.

**Conflicts of Interest:** The authors have no conflict of interest to declare.

**Financial Support:** No financial support was received in the conduct of this study.

## Notes

[1]   The countries constituting the ASEAN-5 are Thailand, Malaysia, Singapore, the Philippines, and Indonesia.

[2]   A more recent study by Basher et al. (2012) does report, however, that positive shocks to oil prices tend to depress emerging market stock prices and U.S. dollar exchange rates in the short run.

[3]   The countries included in the analysis are Tunisia, Morocco, Algeria, Bahrain, Saudi Arabia and Iran (Brini et al. 2016).

[4]   This result is taken as evidence that the efficient market hypothesis (EMH) is inapplicable to south Asian countries.

[5]   In the absence of monthly data on real GDP for India, the industrial production index (IPI) is used measure India's aggregate output.

[6]   This result is consistent with that in Naik and Padhi (2012), which uses 1994–2011 data on industrial production and SENSEX stock prices from India.

[7]   Using 1994–2011 data for India, Naik and Padhi (2012) report bidirectional causality between industrial production and SENSEX stock prices.

[8]   The first of these results is supported by those in a recent study by Przekota and Szczepańska-Przekota (2022).

[9]   According to the results in Table 3, the real exchange rate neither Granger causes, nor is Granger caused by, the price of oil. The latter of these results supports the finding reported in a recent study by Marquez (2022) while both results support some of the findings reported in a recent study by Orzeszko (2021). Neither of these results is supported by those in a recent study by Przekota and Szczepańska-Przekota (2022).

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
