# Peer review of "Causal Relationships between Oil Prices and Key Macroeconomic Variables in India"

_ijfs, doi:10.3390/ijfs11040143_

Round 1

Reviewer 1 Report

Comments and Suggestions for Authors

Although the Introduction section did a great job at introducing the idea, a separate Literature Review section could have been added that could explore similar research studies, perhaps for different set of countries or different methodologies. The Results section could have compared the results with the findings of other papers. Rationale behind the methodologies was explicitly mentioned and easy to understand. Overall, the paper was very reader-friendly.

Comments on the Quality of English Language

Good.

Author Response

Following reviewer 1 's suggestion a separate literature review section has been added, which includes additional literature reviews that include new country sets as well as new methodologies as well.   We thank the reviewer for this suggestion. 

Reviewer 2 Report

Comments and Suggestions for Authors

The paper under review investigates the relationship between oil prices and their impact on the economy. The authors employ a specific data set and statistical methods to explore this connection. However, the study has several significant weaknesses and shortcomings that undermine its validity and robustness.

Major Comments

1. Limited Scope and Data Period: The paper suffers from a notably limited scope, both in terms of the variables considered and the time period analyzed. A broader range of economic indicators and a more extensive timeframe should be incorporated to ensure a comprehensive understanding of the oil price-economic relationship.

2. Assumptions and Simplifications: The authors rely heavily on assumptions and simplifications in their analysis. These assumptions might oversimplify the intricate dynamics of the real-world economic system, leading to inaccurate or skewed conclusions. The base model is a simple regression relating just a few variables. A more nuanced approach, accounting for complexities, is essential.

3. Methodological Limitations: This study's data and methodological approach are not adequately discussed or justified. A clear exposition of the methodology, along with an exploration of potential biases and limitations, is crucial to establish the credibility of the research. Furthermore, the data description is very superficial, not allowing for replicability. The authors do not clarify what type of indexes they use (price/total return), the currency approach, the data sources, how the data is prepared and processed, how RER is estimated, etc.

4. Writing and Structuring: The paper is poorly written and structured. For example, the introduction says nothing about the paper's results and contribution. The reader cannot know what the authors find, why it matters, and how it contributes to literature (and which literature). Lastly, it does not even outline the structure of the paper.

Minor Comments

5. Inadequate Literature Review: The literature review presented in the paper is inadequate and does not comprehensively cover the existing research in the field. A more thorough review of relevant literature is necessary to provide a solid theoretical foundation for the study.

6. Incomplete Policy Implications: The paper lacks a sufficient discussion regarding the policy implications and recommendations derived from the research. Integrating practical policy recommendations is essential to enhance the paper's real-world relevance and applicability.

7. Insufficient Table Descriptions” Tables are not self-contained. The reader cannot understand it without scanning through the entire paper.

Comments on the Quality of English Language

The language is not the biggest problem of this paper.

Author Response

We thank the reviewer for the comments.  We have the following responses to the comments:

  1. Unlike structural modeling, this paper uses a VAR (VECM) method which includes only the relevant variables in the model including their lags. The study covers a 20-year time frame with monthly data of 240 observations. Given the standard practice of time series econometrics, this data set should be enough to run a VECM model.
  2. Even though we start with a functional form of the simple regression model, the final form of the model is a VAR model which is later estimated as VECM after testing for stationarity and cointegration. 
  3. (i) We have used a stock index (SENSEX) instead of one or two stocks.  (ii) data sources are clearly given at the end of the methodology section, RER is the real effective exchange rate that has been derived from the Saint Louis Fed website, which is indicated in the paper.
  4. Following the reviewer's recommendation we have added the significance of our study and the outline of the structure of the paper in the introduction section.
  5. Policy implications are added at the end of the conclusion.
  6. The tables are presented in the standard format, which is self-explanatory for people familiar with econometrics.  For others, they can read the interpretations in the empirical findings section.

Reviewer 3 Report

Comments and Suggestions for Authors

A very good piece of research concerning all the issues described as the objective of the investigation. Sound research regarding the methodology used.

Comments related to a) lines 260-269 and the insufficient interpretation of impulse response analysis, so please expand on this also given additional literature-related studies and b) Conclusions, here there is clearly room for improvement, and there is a need to include more economic substantiation of VECM analysis results other than stating the there are bidirectional causalities. For example, why the VD of aggregated output is so poorly explained by the effect of interest rates, stock prices, or even by oil prices effects? this seems to be strange.

Author Response

We thank the reviewer for the suggestion to expand the interpretation of VDC and the impulse responses. We also want to insist that the focus of the paper is to look at the effect of oil price.  That is why the explanations of other VDCs and impulse responses are not in detail.

Reviewer 4 Report

Comments and Suggestions for Authors

This paper empirically examines the relationship between oil prices and macro variables in India. Although the topic of this paper is undoubtedly important and the econometric methods have been correctly applied, the current paper makes little new contribution to existing research. The empirical method should be reviewed to clarify the contributions of the authors. Individual comments are as follows.

Comment1.

Authors should specify which oil price they used in their analysis. It is also better to specify whether the price is nominal or real, and whether it is denominated in dollars or rupees. Inferences can be made from the data sources, but these should be discussed in the paper.

Comment2

In order to understand the relationship between oil prices and macro variables, it is desirable to include policy interest rates in the analysis.

Comment3

Does Real exchange rate (RER) in the paper mean Real effective exchange rate (REER)? If it is a real exchange rate, you should specify which currency the exchange rate is.

Author Response

  1. The oil price used is the Brent Europe oil price.  It is in US dollars at current price.  This has been added in the data section of the paper.
  2. The focus of this paper is to look at the effect of oil price in major macroeconomic variables.  That is why other macro variables such as interest rates are included.
  3. Yes, it is real effective exchange rate.  Since it is an Indian study, it is the Indian rupee's  real effective exchange rate.

Round 2

Reviewer 2 Report

Comments and Suggestions for Authors

I have read the revised version of the manuscript and stand by my previous comments. The changes made by the authors are largely cosmetic and do not alter my overall assessment of the paper. The empirical analysis does not meet the quality standards of a serious academic journal. Furthermore, the contribution is unclear and the writing is immature. Consider the introduction. It summarizes what the authors do, what they find, and how it contributes to the literature in literally three (sic!) sentences. It is almost impossible to tell how significant the authors' findings are and how they relate to the previous literature. In summary, I maintain my recommendation to reject the submission.

Comments on the Quality of English Language

Overall, ok.

Author Response

  1. This paper uses a Vector Autoregressive Error Correction Model (VECM) which is an appropriate macroeconomic modeling for a research issue that uses time series data (see Walter Anders "Applied Econometric Time Series, Wiley, 3rd Edition, 2010, pp. 297 - 319).
  2. The revised version of the manuscript clearly indicates about the data, their sources and the definition. 
  3. Other comments were incorporated in the revised version of the manuscript.

Reviewer 4 Report

Comments and Suggestions for Authors

With the revise, the authors addressed two of the three comments I mentioned and improved the manuscript considerably. However, since no new analysis has been added, the evaluation on this point remains unchanged. In order to make this manuscript acceptable, the authors need to indicate the position of this paper in relation to previous research and clarify the contribution of this paper.

Author Response

Thank you for your comment on the positioning of the paper.  In order to positioning our paper, we have added some sentences in the introduction section.

Round 3

Reviewer 2 Report

Comments and Suggestions for Authors

I acknowledge that the paper has improved moderately. While I rate the overall contribution and quality as mediocre, the paper could potentially be published. However, I would like to make a few editorial suggestions to strengthen the positioning. 

1. In the introduction, I would recommend separating what the authors do from what the authors find into distinct paragraphs for clarity. In other words, explain the methodology and data in the second paragraph, and then summarize the results in the third paragraph. Finally, the outline of the structure of the manuscript could also be discussed in a separate (last) paragraph of the introduction.

2. The title could be simplified to "Causal Relationships between Oil Prices and Key Macroeconomic Variables in India".

3. The literature review is quite extensive. However, the authors could also refer to the related papers by Naik and Padhi (2012), Basher et al. (2012), or Qian et al. (2023), which examine the drivers of time-series variation in stock returns in emerging markets, including India.

4. The authors might consider dividing section 4 into several smaller subsections with separate subheadings, which would improve the readability of the paper.

5. It would be valuable to comment on the limitations of the paper in the "Summary and Conclusions" section and to indicate areas for further research.

6. Figure 1 is unreadable. The fonts are too small and it is hard to see anything.

7. Authors should make sure that all tables and figures are self-contained, i.e., the reader can understand them without having to scan the entire paper. At present, the notes to the tables are very modest (or missing), so that, for example, the symbols remain unexplained. 

References

Basher, S. A., Haug, A. A., & Sadorsky, P. (2012). Oil prices, exchange rates and emerging stock markets. Energy Economics, 34(1), 227-240.

Naik, P. K., & Padhi, P. (2012). The Impact of Macroeconomic Fundamentals on Stock Prices Revisited: Evidence from Indian Data. Eurasian Journal of Business and Economics, 5(10), 25-44.

Qian, L., Jiang, Y., & Long, H. (2023). What drives the dependence between the Chinese and global stock markets? Modern Finance, 1(1), 12–16. https://doi.org/10.61351/mf.v1i1.5

Author Response

1. In the introduction, I would recommend separating what the authors do from what the authors find into distinct paragraphs for clarity. In other words, explain the methodology and data in the second paragraph, and then summarize the results in the third paragraph. Finally, the outline of the structure of the manuscript could also be discussed in a separate (last) paragraph of the introduction.

We have created distinct paragraphs in the introduction section along the lines suggested above by the reviewer.  We agree that this section is clearer for readers as a result of this change.

2. The title could be simplified to "Causal Relationships between Oil Prices and Key Macroeconomic Variables in India".

We appreciate this comment and have adopted the new paper title suggested by the reviewer.

3. The literature review is quite extensive. However, the authors could also refer to the related papers by Naik and Padhi (2012), Basher et al. (2012), or Qian et al. (2023), which examine the drivers of time-series variation in stock returns in emerging markets, including India.

Rather than just one, as prompted, we were able to make use of two of the three article suggestions provided above by the reviewer.  These are Basher et al. (2012) and Naik and Padhi (2012).  We thank the reviewer for prompting us to examine the possibility of including at least one of these.

4. The authors might consider dividing section 4 into several smaller subsections with separate subheadings, which would improve the readability of the paper.

We divided section 4 into five separate subsections, each with its own subheading.  We agree with the reviewer that this change improves the presentation.

5. It would be valuable to comment on the limitations of the paper in the "Summary and Conclusions" section and to indicate areas for further research.

In response to the reviewer’s comment above, we have added a new paragraph at the end of the paper.  That paragraph is inserted below for your convenience.

Finally, the results presented in this study may not apply globally, or even to other south Asian countries.  Thus, the approach used here for the study of the impact of oil price shocks on India’s economy should be applied to other countries, particularly those in south Asia (e.g., Sri Lanka, Nepal, etc.).  Of course, application of the methods used above for India can also be applied to countries elsewhere on the globe, such as those in eastern Europe and South America.  This line of future research would be particularly beneficial if applied to countries and areas that have been understudied in prior research.

As indicated in the new paragraph above, one limitation of our study is that its results might not apply to other countries.  Thus, we urge researchers to apply our approach to other countries, particularly those in south Asia.

6. Figure 1 is unreadable. The fonts are too small and it is hard to see anything.

Although we don’t dispute the reviewer’s comment above, Figure 1 is collapsible/expandable.  Thus, its size can be addressed at the production stage.

7. Authors should make sure that all tables and figures are self-contained, i.e., the reader can understand them without having to scan the entire paper. At present, the notes to the tables are very modest (or missing), so that, for example, the symbols remain unexplained.

We agree with the reviewer’s point above.  To address it, we have added a key to the footnotes of Table 1, Table 3, and Table 4.  In each case the new key defines the variables included in the tables.

On behalf of my coauthors, I thank Reviewer 2 for his or her careful attention to our work, and for the above comments which helped us improve the manuscript.